# Total Quadriceps Resection in High-Grade Soft-Tissue Sarcomas of the Thigh: Surgical Technique and Long-Term Functional Outcomes in Surviving Patients

**DOI:** 10.3390/cancers18010037

**Published:** 2025-12-22

**Authors:** Luis Rafael Ramos Pascua, Paula Casas Ramos, Rubén Álvarez García, Sergio Sánchez Herráez, Cristina Ojeda Thies, Maximiliano Eugenio Negri, Daniel Bustamante Recuenco, Jesús Enrique Vilá Rico

**Affiliations:** 1Department of Trauma and Orthopaedics Surgery, University Hospital 12 de Octubre, Av/Córdoba, s/n, Usera, 28041 Madrid, Spain; ojedathies@gmail.com (C.O.T.); danibustamante@telefonica.net (D.B.R.); vilayrico@gmail.com (J.E.V.R.); 2School of Medicine, Complutense University, Pl. de Ramón y Cajal, s/n, Moncloa-Aravaca, 28040 Madrid, Spain; 3Department of Trauma and Orthopaedic Surgery, IOTAM, Quirónsalud University Hospital, C/Diego de Velázquez, 1, Pozuelo de Alarcón, 28223 Madrid, Spain; 4Department of Trauma and Orthopaedic Surgery, University Hospital of León, Calle Altos de Nava, s/n, 24008 León, Spain; casaspaula@hotmail.com (P.C.R.); herraezsergios@yahoo.es (S.S.H.); 5Department of Plastic and Reconstructive Surgery, University Hospital of León, Calle Altos de Nava, s/n, 24008 León, Spain; rualvarezgarcia@gmail.com; 6Department of Plastic and Reconstructive Surgery, HM La Esperanza Hospital, Avenida das Burgas, 2, 15705 Santiago de Compostela, A Coruña, Spain; 7Department of Trauma and Orthopaedic Surgery, Virgen de la Peña General Hospital of Fuerteventura, Carretera del Aeropuerto, Km 1, 35600 Puerto del Rosario, Las Palmas, Spain; 8Department of Trauma and Orthopaedic Surgery, Hospital Universitario Ruber Juan Bravo University Hospital, Calle de Juan Bravo, 49, 28006 Madrid, Spain

**Keywords:** soft-tissue sarcoma, anterior compartment of the thigh, quadriceps, quadricectomy, local muscle transfers, extensor mechanism of the knee, treatment, prognosis

## Abstract

One challenge of compartmental resection of soft-tissue sarcomas of the anterior thigh is the disruption of the quadriceps extensor mechanism. The authors present their results in 10 patients followed postoperatively for a minimum of 4 years in the 5 surviving patients who underwent conservative surgical procedures and discuss their surgical technique to reestablish extensor function of the knee. Local muscle transfers are more suited for low-demand patients, while neurotized free muscle flaps are mainly an option for young, motivated patients. Resection appears to be technically easier if performed distally to proximally in the thigh.

## 1. Introduction

Soft-tissue sarcomas are most commonly located in the thigh [1,2], with this location accounting for approximately one third of total cases [3]. Most are deep and situated in the anterior compartment [3]. The femur is usually not infiltrated by the tumor in these cases due to its periosteal envelope and, occasionally, the interposition of the vastus intermedius muscle. General treatment does not differ from that recommended for soft-tissue sarcomas in other locations [4], although the anatomy and function of the quadriceps muscle entails some surgical peculiarities for reconstruction.

Wide-margin resection of soft-tissue sarcomas of the anterior compartment of the thigh may require sacrificing most of the quadriceps and, consequently, cause a significant extension deficit and loss of stability of the knee. Resection of a greater amount of quadriceps muscle groups leads to worse outcomes; good results can be expected if at least two groups are preserved [5], but this is not always feasible.

The reconstructive strategy—especially in cases with complete muscle resection—is equivalent to situations in which the femoral nerve has been sacrificed or damaged, and it aims to cover the soft-tissue defect and recover extensor function of the knee. Surgical strategies include local tendon transfers, allograft reconstruction, composite tissue flaps, and free muscle flaps; the best reconstruction technique remains to be defined, and evidence is limited to brief case series [4,5,6,7,8,9,10,11,12,13,14,15,16,17]. Among local tendon transfers, the technique classically described involves transfer of the sartorius or biceps femoris, combined or not with transfer of the semitendinosus or gracilis muscles [6,16,18,19]. Most patients will require walking aids following these transfers, and complication rates are high, particularly for combined tendon transfers [5]. Gait recovery may be more favorable following free muscle transfers such as latissimus dorsi flaps, which require microsurgical capabilities that are not universally available [4,7,8].

The aim of this study is to present our outcomes after a minimum 4-year follow-up in surviving patients with limb preservation with high-grade soft-tissue sarcomas who required total quadricectomies followed by quadriceps reconstruction.

## 2. Materials and Methods

We performed a retrospective review of patients treated for AJCC stage IIIB high-grade soft-tissue sarcomas of the anterior compartment of the thigh treated in two hospitals, all but one by the same surgeon, from 2012 to 2025. Consecutive patients with the aforementioned diagnosis who underwent total quadriceps resection and suffered complete loss of the knee extensor mechanism were included in the study. Patients were excluded if longitudinal continuity of any of the quadriceps muscle complex was preserved. During follow-up, there was no loss of patients or of any data required for the study.

The study was approved by the Ethics Committee (CEIm) of our Center, complying with patient anonymity requirements. All survivors provided written informed consent for publication. All surviving patients provided written informed consent to be included in the case series. For non-surviving patients, data were irreversibly anonymized. All procedures were performed in accordance with the Helsinki Declaration of 1975, as revised in 1983, as well as local legislation (Law 41/2002 regulating patient autonomy and the rights and obligations regarding information and clinical documentation).

Ten patients (four male and six female) were included, with a mean age of 58.4 years (range: 35–79 years). The mean follow-up of surviving patients (5 cases) until their death (4 cases) or the performance of an external hemipelvectomy due to some type of complication (1 case) was 65.9 months (range: 5 days–163 months). The 5 surviving patients with limb preservation were followed up for an average of 109.8 months (range: 51–163 months).

Demographic, clinical, and oncologic data regarding tumor staging is included in Table 1. All patients presented with a progressively growing mass of less than 10 months duration (except for case 9, who consulted over a year after initiating symptoms) and discomfort of variable intensity. Mean craniocaudal length in magnetic resonance imaging was 14.5 cm (range: 9.4–25 cm) (Figure 1B and Figure 2B). Seven patients had undifferentiated pleomorphic sarcomas, two had myxofibrosarcomas, and another a clear cell sarcoma. In one patient (case 4), the tumor was initially mistaken for a posttraumatic hematoma, which had been drained via an incision though the *vastus lateralis* muscle. All patients were treated following standard clinical guidelines [20,21]. Closed core needle biopsy via an anteromedial or anterolateral approach was performed in seven cases. Biopsy was performed twice in case 7. Case 4 had an open biopsy during drainage of the supposed hematoma. One patient (case 6) was referred from another center after a biopsy performed using three different approaches on the anterior aspect of the thigh (Figure 2A).

All patients underwent total quadriceps resections that were supervised by the same surgeon in all cases (LRRP). Reconstruction was performed in eight patients using local tendon transfers with a modification of the technique originally described by Sugarbaker. One patient (case 6) with a massive loss of skin and muscle of the anterior compartment of the thigh after tumor resection had reconstruction performed with a free functional muscle transfer from the contralateral thigh. This was decided upon after discussing the different reconstructive options with the patient due to her younger age and very active lifestyle. A second older patient (case 8) was treated with the same surgical technique due to the good results obtained in case 6. Adjuvant therapies are detailed in Table 1. Three patients received postoperative radiotherapy, and three had postoperative chemotherapy. Case 6 was subjected to neoadjuvant radiotherapy and chemotherapy, after which re-staging was performed, followed by postoperative adjuvant chemotherapy. Case 10 also received neoadjuvant radiotherapy and neoadjuvant chemotherapy, but treatment was discontinued due to local tumor progression.

### Surgical Technique

With the patient in the supine position and the affected limb free to allow for flexion and abduction/adduction of the hip, a long curved longitudinal incision was made from the antero-superior iliac spine to the anterior tibial tuberosity, with an ellipse around the biopsy tract (Figure 3A). Both flaps were dissected superficial to the deep fascia of the thigh until the projection of the intermuscular septa [8]. The fascia was incised over the *sartorius* muscle, and, using this muscle as a reference and with the possibility of it being included in the resection bloc, the femoral neurovascular bundle was identified and referenced with a cotton or silicone loop (Figure 3B). After releasing Hunter’s canal (or the adductor canal), the quadricipital tendon was dissected and cut two finger-breadths (approx. 3 cm) above the patella. The quadriceps was released subperiostically from the femur, from distal to proximal, by pulling the distal end of the quadriceps muscle proximally, while elevating the periosteum (included in the resection bloc) with a periosteotome (Figure 3C) [22]. The branches of the deep femoral artery were identified at the level of the lesser trochanter, at the latest, and were ligated as found while elevating the quadriceps. The resection was completed with manual reference to the sarcoma, including the necessary myotomies of the quadriceps, and the resection bloc included the proximal ends of the *vastus intermedius* and *rectus femoris* muscles, i.e., the entire quadriceps.

Reconstruction using local musculo-tendinous transfers started by localizing the lateral head of the *biceps femoris* laterally and the *sartorius* muscle medially, or, if this muscle has been sacrificed, the *gracilis* muscle. Their distal tendinous portions were cut, liberated proximally, and sutured to the remnant of the quadriceps tendon and among themselves, and, if possible, to the patella itself [6], allowing for coverage of the distal 10–15 cm of the femur (Figure 1 and Figure 3D). Two or three suction drains were placed in the wound, and closure was completed after trimming the skin flaps to adapt them to the defect.

For cases in which a reinnervated free muscle transfer flap was preferred, the flap was obtained from the contralateral thigh, using a myocutaneous flap including 16 cm of the *vastus lateralis* muscle (Figure 2 and Figure 4). The muscle was marked every 5 cm to maintain constant tension at the time of insertion. The descending branch of the lateral circumflex femoral artery was preserved and used as a recipient vessel (Figure 5). The motor branch of the vastus lateralis muscle flap was coapted to the femoral nerve stump. Next, the flap was sutured to the residual stumps of the quadriceps resection. The *sartorius* muscle from the ipsilateral thigh was also transferred to augment the knee extensor mechanism.

A compressive bandage was applied upon completion of all surgeries, and a posterior extension splint was applied for 1–3 weeks, which was subsequently replaced by an articulated knee brace. This orthosis was worn for 3 months. Early postoperative mobilization out of bed and active ankle mobility were encouraged. Drains were removed once they became unproductive, usually not before one week after surgery. Partial weight bearing of the operated limb was permitted after 1–3 weeks. Sutures were removed 2–4 weeks after surgery. Upon suture removal, full weight bearing as tolerated was permitted with the aid of crutches, which were maintained for at least 3 months. Formal physiotherapy usually did not start until one or two months postoperatively, conditioned by the schedule of the adjuvant therapies. Case 6 continued physiotherapy for over one year. No patient used the dorsiflexion blocking orthosis proposed by Sugarbaker and Lampert [18] to prevent flexion of the knee upon contact with the floor.

Surgical complications (e.g., hematoma, infection, and wound dehiscence) and functional and oncological results of treatment were analyzed at the final follow-up or death. Functional results were evaluated according to the Musculoskeletal Tumor Society (MSTS) scale [23] in 7 of 10 patients who survived more than 3 years at the end of the study or at the last review before death (cases 1 and 9). The MSTS scale converts the score allocated to six criteria into percentages, and considers the results above 79% good or excellent, between 60 and 79% acceptable, and below 60% poor. Strength was graded from 0 to 5 according to the Medical Research Council Manual Muscle Testing scale [24]. To avoid assessor bias of the primary operating surgeon, we used the values recorded by a specialist in physical therapy and rehabilitative medicine. Oncological results were evaluated in terms of local recurrence, metastasis, and death. Continuous variables are summarized using the average values, standard deviations, and ranges. Categorical values are indicated using percentages. No formal statistical comparisons were made due to the nature of the study design.

## 3. Results

Wide resection margins were achieved in all ten cases, with at least fascia and periosteum interposed between the tumor and the plane of surgical resection. Four patients died during follow-up; one during the immediate postoperative period due to a massive pulmonary embolism (case 3), another died eleven months after surgery due to general multiorgan failure following surgical complications (case 5), and two due to metastatic disease progression at 50 months (case 1) and 43 months (case 9).

Relevant surgical complications appeared in eight cases (80%). The superficial femoral artery was accidentally torn in one patient intraoperatively (case 1) and was repaired without incident. Six patients suffered necrosis or dehiscence of the wound margins. Five of them were older than 65 years. One patient (case 5), a 79-year-old female, had to have an amputation due to widespread necrosis of the surgical wound edges and infection, and later died at 11 months of follow-up due to multisystemic failure precipitated by the surgeries and treatment. Another four patients required debridement and primary closure four to eight weeks after initial anterior compartment resection due to varying degrees of necrosis of the wound margins. A V.A.C. (Vacuum Assisted Closure) Therapy System was placed in one patient (case 8). This patient had a quadriceps tendon rupture a year and a half later as a result of a fall and forced flexion of the knee. She was treated with direct sutures and reinforcement with an Achilles tendon allograft. Another patient developed a seroma, which was repeatedly evacuated under sterile conditions over a three-month period (case 7). This patient was initially suspected to have a surgical site infection and was readmitted one month after surgery with fever and neutropenia. Cultures of the aspirated material were negative, and empiric antibiotic therapy was prescribed; once the seroma had been cleared, no signs of infection were observed. Case 10 had a local recurrence of sarcoma and underwent a modified external hemipelvectomy with a posterior flap 6 months after the first surgery.

The functional results of the five patients who remained alive and retained their limb after a minimum four-year follow-up (four of them with a minimum follow-up of 8 years) were good or excellent in two cases, acceptable in one, and poor in two (Table 2). Case 9, who survived for almost 4 years, had a good functional outcome until his death (26/30 MSTS score). The five surviving patients expressed satisfaction with the outcome of the treatment. Average knee flexion in these patients was 80° (range: 10–150°). Passive extension was complete in all cases, though no patient achieved it actively. Extensor strength was 2/5 in four patients and 4/5 in the other (case 6).

Regarding the oncologic results of the patients that survived more than 12 months after surgery, one patient showed local recurrence and underwent a hemipelvectomy (case 9); one patient died 50 months after the quadriceps resection after developing metastases after six months (case 1), and another developed late metastases and died 43 months after surgery. In summary, five patients were alive and disease-free and retained a functional limb at the final follow-up.

## 4. Discussion

Limb-sparing surgery has been the accepted method of treatment for most patients with bony and soft-tissue sarcomas since the 1980s, including for the majority of patients with high-grade lesions [25]. This involves resection with oncologic margins and functional reconstruction. The quality of the surgical margins is the strongest prognostic factor of local outcomes when treating soft-tissue sarcomas [26] and apparently correlates with the appearance of late metastases [27]. In high-grade sarcomas, the oncologic margins must be wide or radical (intra- or extracompartmental resections, respectively), with healthy tissue between the tumor and the surgical plane of resection [20,21].

The quadriceps is included in the anterior compartment of the thigh, limited by fascias and the periosteum that surrounds the femur. These structures are anatomical barriers that resist tumoral growth [18,20,28], although histological bone and isolated periosteal infiltration in 49% and 14.3% of patients, respectively, in a series of 49 deep soft-tissue sarcomas in contact with the bone have been reported [29]. Therefore, large high-grade soft-tissue sarcomas that do not cross these barriers—even if contacting them—can be resected with oncologic margins by en-bloc removal of the quadriceps and its envelope, even if the cranial and caudal muscle insertions are not completely removed. The absence of local recurrences in most cases in our series corroborates this. Unlike the classical technique [8,18], resection seems easier if the distal quadriceps tenotomy is performed first and the excision is continued by pulling the muscle proximally. The *sartorius* muscle, considered by some as part of the anterior compartment of the thigh [30], can be considered individually, as can the *gracilis* muscle.

Kulayat and Karakousis [22] proposed a modified technique for resection of large soft-tissue sarcomas of the anterior compartment of the thigh that preserved one of the quadriceps’ heads, usually the vastus medialis, with intact innervation as its nerve branch remains independent and external to the muscle until the lower half of the vastus medialis. The authors spoke of a ‘quasi-compartmental resection’ that facilitates recovery, allowing their patients to have full and unimpaired function of the extremity with satisfactory local control of the disease. Karakousis [31], with the same conservative philosophy, had previously theorized that if a sarcoma is located near the proximal insertion of the quadriceps muscle, it did not make much sense to attempt a 20–30 cm margin in the opposite direction, and if it was located in the vastus lateralis, it did not make sense to remove the vastus medialis. Sadly, these options were not safely possible in any of our cases.

The aim of reconstruction following quadriceps resection is threefold: aesthetic, defect coverage, and functional recovery. Currently, consensus is lacking regarding the best method of reconstruction, and controversy continues (Table 3). In fact, as Fisher et al. [6] published, the selection of the surgical technique was not associated with the extent of quadriceps muscle resection but with the preference and experience of the surgeon. Muscle transfers, originally described for the treatment of quadriceps palsy due to poliomyelitis [32], are usually the procedures of choice, as they are technically quick and easy to perform, with relatively little donor site morbidity. On the medial side of the thigh, any of the muscles of the *pes anserinus* can be used, preferably the *sartorius* muscle, although, as it is innervated by the femoral nerve, it would not be functional if the nerve has been sacrificed during resection. Laterally, the lateral head of the biceps muscle is preferably chosen for reconstruction.

Pritsch et al. [19] published the clinical results of their series of 15 patients in whom the extensor mechanism of the knee had been reconstructed using transfers from the sartorius, semitendinosus, and femoral biceps muscles after resection of soft-tissue tumors of the anterior compartment of the thigh. The functional outcomes in these patients were good or excellent in 13 (86.7%) cases, with a satisfactory range of active flexion that correlated with the extent of quadriceps resection. The quadriceps had been completely removed—as in our series—in only one of the fifteen cases, and the outcome was considered as only fair (grading the knee extension strength as 3/5).

Fischer et al. [6] described the outcomes of a series of 43 patients treated with local muscle transfers; 16 had had complete quadriceps resections—as in our series—and complication rates increased with complete (67%) vs. partial resections (13% complication rate), as well as with combined transfers vs. sole *biceps femoris* transfers. Functional outcomes were also worse in patients with complete resections, with 78% requiring walking aids.

Despite the high rate of local wound complications in our series, the technique described is simple to perform and yields acceptable functional results. However, active extension and strength were significantly reduced, and this compromised knee stability. For all these reasons, the procedure could be more suitable for partial quadriceps resections [5] or in older, low-demand patients or those with comorbidities not fit for prolonged microsurgical reconstruction. Nevertheless, particularly in older patients and according to our findings, postoperative necrosis of the surgical wound edges appears to be more frequent and should be taken into consideration. Similar results were reported in the series by Fisher et al. [6]. In their study of 43 patients undergoing hamstring transfers, postoperative complications occurred in sixteen patients (37%). Wound dehiscence was seen in 10 (62%), lymphedema in 2 (12%), fistula in 1 (6%) and hematoma in 1 patient (6%). Additionally, two patients developed both wound dehiscence and lymphedema (12%).

Free functional muscle transfer has been reported to restore muscle function of the lower extremity, but there is less data available using this technique [4,5,6,7,8,9,10,11,12,13,14,15,16,17,33,34,35,36,37]. Reconstruction of the extensor mechanism of the knee can be achieved with free muscle transfers of the *latissimus dorsi*, *gracilis*, *tensor fascia latae*, *rectus abdominis*, or *vastus lateralis* muscles [36]. The selected muscle should have an adequate shape, strength, and range of movement to adapt to the receptor area. More voluminous muscles will offer a more potent contraction, as muscle contraction is proportional to the muscle’s cross-sectional area [38].

Ihara et al. [37] reported on 23 patients treated with neurotized free muscle flaps after resection of sarcomas of the extremities. The latissimus dorsi muscle was used to reconstruct large defects of the quadriceps in eight cases, with satisfactory results. Hallock et al. [39] and Wilcox et al. [40] also published case reports with favorable results using *latissmus dorsi* transfers as a reconstruction method. Innocenti et al. [8] reviewed a series of 11 patients with soft-tissue sarcomas treated with partial or total resection of the quadriceps, followed by reconstruction using a vascularized free *latissimus dorsi* flap, and adjuvant chemo- and/or radiotherapy. They observed that this type of reconstruction, combined with *sartorius* transfer, contributed to knee stability and offered a certain degree of active extension, enough to allow for unassisted ambulation. The speed of recovery was affected by patient age, the distance between the nerve and the muscle, the extent of administered radiotherapy, and intraoperative ischemia of the flap. Grinsell et al. [35] reported 11 successful functional reconstructions using *rectus abdominis* myocutaneous flap cases. Walley et al. [17] and Stranix et al. [12] shared their experiences using flaps from the contralateral thigh, as in our case. All patients achieved ambulation, and MSTS scores ranged from 19 to 24. Full passive extension was recovered in all patients.

The *vastus lateralis* is the largest and strongest muscle of the quadriceps, making up 45% of its mass and acting as the greatest muscle stabilizer of the knee. Compared to the *latissimus dorsi* flap, the *vastus lateralis* anterolateral thigh flap has the advantage of providing a functional muscle with the same native vectors as the resected muscle, reestablishing natural knee extension kinematics. The low morbidity associated with harvesting the *vastus lateralis* muscle has been widely documented [41]. Compared with functional transfers of the latissimus dorsi or gracilis muscles, whose muscle fibers are distributed in a parallel fashion, the vastus lateralis muscle has a pennate configuration, at an angle to the force-generating axis (pennation angle). Pennate muscles offer shorter fiber lengths and displacements, but on the other hand, they contain more fibers per unit of muscle volume and can generate more force than parallel-fibered muscles of the same size [38]. The cases described in our series confirm the good results attributed to the technique and low donor-site morbidity, showing that the contralateral *vastus lateralis* muscle can be an alternative to classic muscle transfers.

Regarding adjuvant therapies in the management of high-grade soft-tissue sarcomas of the extremities, radiotherapy is a mainstay in the majority of protocols, while chemotherapy is reserved for non-elderly patients [27,42]. A comparative study of the results of the management of soft-tissue sarcomas of the thigh concluded that there were distinct patterns of morbidity depending on the anatomical compartment of origin; for example, wound dehiscence and edema were more likely in the medial compartment. On the other hand, only the use of external-beam radiation therapy and a tumor size of more than 10 cm were independent predictors of joint stiffness [2]. Sugarbaker and Lambert [18] also warned of knee stiffness due to the delivery of high-dose radiation therapy to the entire knee joint as required in such a large approach.

Our study has several limitations. First, we include a small number of cases, with only two managed with free muscle transfer from the contralateral thigh. However, we do consider that our case series, specifically describing total quadricectomies and with a number in line with those published by several other authors, provides further knowledge and understanding of this subject. Second, oncologic and functional follow-up was reduced to seven cases due to the death of two patients in the early postoperative period and early local recurrence that necessitated amputation in another case; nonetheless, all patients that were alive at the time of minimum follow-up could be evaluated and included in this series. Therefore, the survival bias that may exist with regard to functional outcomes is not relevant to the analysis of long-term limb functionality. It is also irrelevant to record the degree of knee flexion in patients with a survival rate of less than one year (cases 3 and 5) or who have undergone an amputation (case 10). Third, there is a risk for co-treatment bias, as long-term treatment-related complications are possible in all cases beyond the minimum 8-year follow-up. Specifically, the location of the tumor in the anterior compartment, the periosteal stripping of the femur, and radiotherapy could potentially lead to pathological femoral fractures, which can occur more than 10 years after treatment [30,43]. Despite these limitations, we believe our study fulfills its proposed objectives. The participation of a single surgeon in all surgeries supports the reliability of the results observed in this series. Furthermore, although the study design was retrospective, there was no loss of patients during follow-up, and all necessary data were recorded in the medical records.

## 5. Conclusions

Total quadriceps resection in large high-grade soft-tissue sarcomas of the anterior compartment of the thigh ensures wide resection margins and local control of the disease, although local wound complications are common, particularly in older patients. Resection appears to be technically easier if performed distally to proximally in the thigh. Reconstruction can be performed using local muscle transfers in low-demand patients, while free neurotized muscle flaps are advisable in selected patients, mainly motivated young people. The anterolateral flap of the *vastus lateralis* muscle of the thigh is a good alternative to the *latissimus dorsi* flap, providing similar versatility while avoiding donor-site morbidity associated with harvesting a major muscle of the upper back.

## Figures and Tables

**Figure 1 cancers-18-00037-f001:**
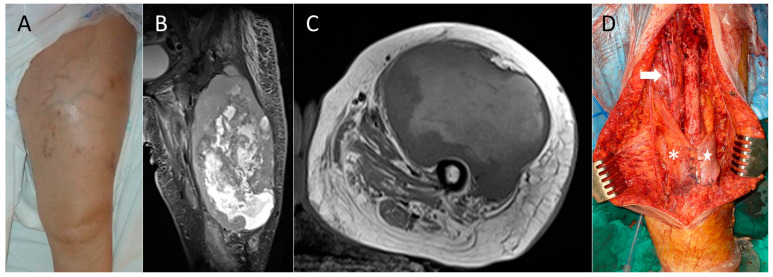
Case 5, reconstructed with local muscle transfer. (**A**) Clinical aspect of the thigh. Note the increased superficial venous circulation. (**B**) T2-weighted magnetic resonance image, coronal view. (**C**) T1-weighted magnetic resonance image, axial view. (**D**) Transfer of the gracilis muscle (asterisk) and the lateral head of the femoral biceps muscle (star). The arrow marks the femoral neurovascular bundle.

**Figure 2 cancers-18-00037-f002:**
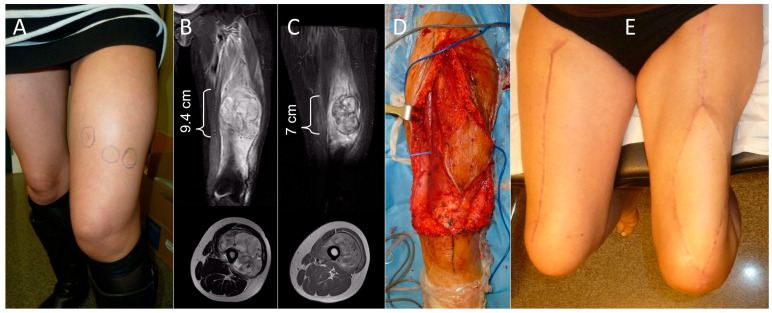
Case 6. (**A**) Clinical aspect of the thigh. Note the three incorrect biopsy approaches marked with circles on the skin, performed at the center of origin from which the patient was referred to us. (**B**) T1- and T2-weighted magnetic resonance images with coronal and axial views before neoadjuvant chemotherapy. (**C**) T1- and T2-weighted magnetic resonance images with coronal and axial views after neoadjuvant chemotherapy. (**D**) Microsurgical transfer of a reinnervated flap of the contralateral vastus lateralis muscle. (**E**) Clinical aspect of both thighs at one-year follow-up.

**Figure 3 cancers-18-00037-f003:**
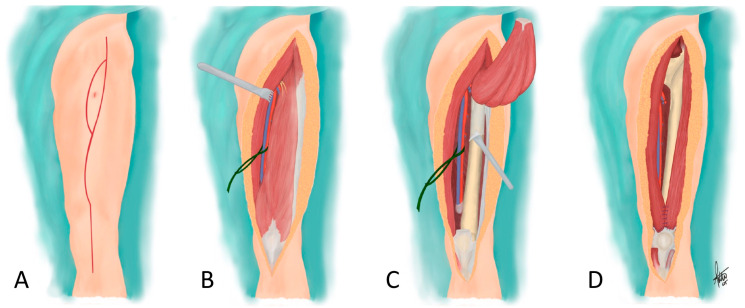
Schematic representation of the surgical technique of anterior compartment resection. (**A**) The limb is draped to allow for free mobilization of the hip; a long curved longitudinal incision is made from the antero-superior iliac spine to the anterior tibial tuberosity, with an ellipse around the biopsy tract. (**B**) Both flaps are dissected up to the projections of the intermuscular septa of the anterior compartment; the femoral neurovascular bundle is identified following the sartorius muscle and referenced. (**C**) The quadricipital tendon is dissected and cut one inch above the patella, and the distal end of the quadriceps muscle is pulled proximally as it is released subperiostically from the femur with a periosteotome, including the periosteum in the resection bloc. (**D**) Reconstruction is performed using local musculo-tendinous transfer of the lateral head of the *biceps femoris* laterally and the *sartorius* muscle medially by cutting and suturing their distal tendons to the remnant of the quadriceps tendon and among themselves, allowing for coverage of the distal third of the femur.

**Figure 4 cancers-18-00037-f004:**
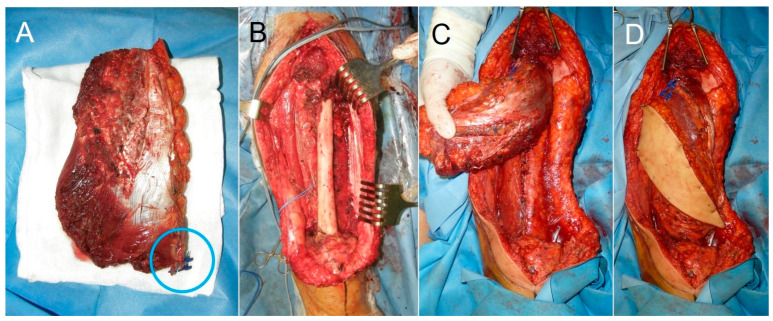
Case 6. (**A**) Flap of the contralateral vastus lateralis muscle with the neurovascular pedicle (blue circle). (**B**) Total quadriceps resection. (**C**,**D**) Flap of the contralateral vastus lateralis muscle transferred to the receptor bed.

**Figure 5 cancers-18-00037-f005:**
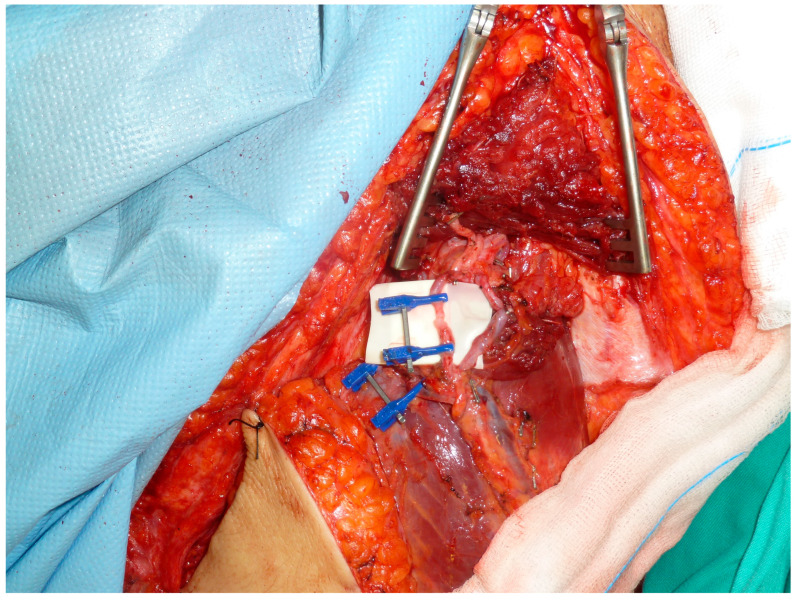
Case 6. Detail of microvascular reconstruction.

**Table 1 cancers-18-00037-t001:** Summary of the cases included in the series. Abbreviations: MRI = Magnetic resonance imaging (* dimensions in the craniocaudal, anteroposterior, and transverse planes). UPS = Undifferentiated pleomorphic sarcoma. MFS = Myxofibrosarcoma. CCS = Clear cell sarcoma.

Case	Age	Sex	Occupation	MRI Size (cm *)	Histologic Type	AJCC Stage	Type of Reconstruction	Radiotherapy	Chemotherapy
1	65	M	Self-employed	15 × 9.5 × 6.5	UPS	IIIB	Local muscle transfer	Postoperative	Postoperative
2	66	M	Self-employed	14 × 8 × 19.5	UPS	IIIB	Local muscle transfer	Postoperative	Postoperative
3	53	F	Unemployed	14 × 11 × 6.3	UPS	IIIB	Local muscle transfer	No	No
4	74	F	Retired	18 × 6 × 13	UPS	IIIB	Local muscle transfer	Postoperative	No
5	79	M	Retired	25 × 10 × 13	UPS	IIIB	Local muscle transfer	No	No
6	35	M	Judge	9.4 × 4.2 × 9.4	MFS	IIIB	Free neurotized muscle flap	Preoperative	Pre- and Postoperative
7	46	F	Businessman	10.6 × 9.6 × 12.5	UPS	IIIB	Local muscle transfer	No	Postoperative
8	54	F	Unemployed	12 × 9.5 × 6.5	MFS	IIIB	Free neurotized muscle flap	Preoperative	Preoperative
9	48	F	Businessman	16 × 10 × 12	CCS	IIIB	Local muscle transfer	No	No
10	64	F	Unemployed	11 × 7.4 × 7.5	UPS	IIIB	Local muscle transfer	Preoperative	Preoperative

**Table 2 cancers-18-00037-t002:** Summary of the outcomes of the patients included in the series. MSTS results [16]: Pain–Function–Emotional acceptance–Supports–Walking–Gait. * Functional results one year after surgery.

Case	Complications	Follow-Up (Months)	Local Recurrence	Metastases	Death	MSTS Outcome [5]	Flexion (Degrees)
1 *	Wound edge necrosis	50	No	Yes	Yes	4-2-4-1-3-2 = 16 (53.33%)	20
2	Wound edge necrosis	163	No	No	No	5-3-5-1-3-2 = 19 (63.33%)	10
3	Death	5 days	No	No	Yes	-	-
4	None	124	No	No	No	5-3-5-4-3-4 = 24 (80%)	140
5	Wound edge necrosis, infection	11	No	No	Yes	-	-
6	None	115	No	No	No	5-5-5-5-5-5 = 30 (100%)	150
7	Seroma	96	No	No	No	5-3-5-4-4-4 = 25 (83.33%)	130
8	Wound dehiscence, quadriceps tendon rupture	51	No	No	No	4-3-4-1-3-2 = 17 (56.67%)	90
9 *	Wound edge necrosis	43	No	Yes	Yes	5-3-5-5-4-4 = 26 (86.67%)	140
10	Wound edge necrosis	6	Yes	No	No	-	-

**Table 3 cancers-18-00037-t003:** List of relevant publications on the reconstruction method following complete quadriceps resection and its functional outcomes. The selection was obtained through an electronic and manual search in PubMed using the search string “soft tissue sarcoma AND quadriceps AND extensor AND reconstruction”. Among the 10 articles identified in the search, we excluded one referring to bone osteosarcomas reconstructed with megaprostheses [15], another on an approach to the distal femur through the knee extensor mechanism [9], another on a new technique of patellar ligament reconstruction [14], another on rotationplasty (Borggreve–Van Nes operation) [13], another that was an evidence-based review [10], and a fifth that was a letter to the editor [16]. This was replaced by the original article by Innocenti et al. [8], which it comments on. Three secondary references found via bibliographic linkage [7,12,19] were added.

Authors	*n*	Age (Average, Years)	Histologic Type	Knee Extensor Functional Reconstruction	Follow-Up (Average, Months)	Postoperative General Complications (Non-Oncological)	Functional Results (Average, Different Scales)	ROM (Average, Extension–Flexion)
Walley et al., 2017 [17]	3	53.6	MFS, LS, OS (distal femur)	Contralateral free ALT and combined muscle (vastus lateralis) or fascia (crural fascia) flaps	20.5	Multiple falls and direct blows (*n* = 1), wound breakdown and distal femur fracture from a fall requiring surgery (*n* = 1)	MSTS score: 72%	ROM: −18°–88°
Müller et al., 2018 [11]	6 ^a^	NS	MFS (*n* = 2) UPS (*n* = 2), SS (*n* = 2)	Fresh-frozen allograft	80.4	Revision surgery (*n* = 2, 1 removal of the allograft)	ISOLS score: 24.7	ROM: −10°–82.5°
Muramatsu et al., 2010 [10]	1	69	MFS	Free vascularized latissimus dorsi myocutaneous flap	18	No	Strength of active knee extension: 4+ of 5 in the muscle manual test	ROM: −10°–110°
Innocenti et al., 2009 [8]	4 ^b^	37.5	LS (*n* = 2), FS (*n* = 2)	Free functioning latissimus dorsi muscle flap from the same side ^c^	69 ^d^	Infection (*n* = 1), late femoral fractures by RT (*n* = 5), secondary operations (*n* = 10) ^d^	MSTS score: 73% excellent or good ^d^	NS
Fischer et al., 2015 [6]	43 ^e^	59.2	NS	Hamstring transfer: sole biceps (n = 31) and biceps + semitendinosus or gracilis (n = 12)	61	Wound dehiscence and/or hematoma (n = 13), lymphedema (n = 2), fistula (n = 1), Revision surgery (n = 7)	QoL: 74% Karnofsky Performance Scale Index: 78% ^f^	ROM (Flexion mean, % of healthy side): 60% ^f^
Houdek et al., 2021 [7]	2 ^g^	51	UPS (n = 2)	Free functional latissimus flaps	24 (mean clinical FU)	NS. Hematoma (*n* = 2), delayed wound healing (*n* = 2), lymphedema (*n* = 1), deep venous thrombosis (*n* = 1). Revision surgery (*n* = 2) ^h^	MSTS score: 70–97%	ROM: 95°
Stranix et al., 2018 [12]	1	73	LS	ALT flap	36	No	MSTS: 77%	NS
Pritsch et al., 2007 [19]	1 ^i^	70	NS	Functional muscle transfer: long head of the biceps femoris muscle, sartorius muscle, and semitendinosus muscle	NS	NS	MSTS: Fair	ROM: −20°–90°
Ramos et al, 2025 [this series]	10	58.4	UPS (*n* = 7) MFS = 2 CCS = 1	Functional muscle transfer: long head of the biceps femoris muscle, sartorius muscle, and/or gracilis muscle (*n* = 8). ALT flap: Myocutaneous flap of vastus lateralis muscle (*n* = 2).	48 ^j^	Wound edge necrosis (*n* = 6), massive pulmonary embolism and death (*n* = 1), infection (*n* = 1), seroma (*n* = 1), quadriceps tendon rupture (*n* = 1)	MSTS score: 76.66% ^k^	Flexion ^k^: 104

ALT = Anterolateral thigh. CCS = Clear cell sarcoma. ISOLS = International Society of Limb Salvage. FS = Fibrosarcoma. FU = Follow-up. LS = Liposarcoma. MFS = Myxofibrosarcoma. MSTS = Musculoskeletal Tumor Society. *n* = Number of cases. NS = Not specified. OS = Osteosarcoma. QoL = Quality of life (SF-36 questionnaire). ROM = Range of motion (degrees). RT = Radiotherapy. SS = Synovial sarcoma. UPS = Undifferentiated pleomorphic sarcoma. ^a^ Resection around the knee joint (quadriceps tendon, patella, and/or patellar tendon) without apparent joint involvement. ^b^ Total quadriceps resection in 4 cases (11 cases in total). ^c^ Associated surgery: sartorius transfer and massive allograft (1 case) or pedicled fascia lata (2 cases). ^d^ Calculated on 11 patients. ^e^ Total quadriceps resection in 16 cases (37%), three fourths in 12 cases (28%, vastus lateralis or medialis plus intermedius and rectus femoris muscle), half in 4 cases (9%, vastus lateralis/medialis and rectus femoris muscle), and the femoral nerve in 11 cases (26%). So, in 91% of cases, 3/4 or more of the quadriceps muscle was removed. ^f^ Results for 9 patients available for physical examination with resection > 3/4 of the quadriceps. ^g^ Total quadriceps resection in 2 patients (12 cases in total). ^h^ Calculated on the 12 patients in the series. ^i^ Total quadriceps resection in 1 patient with resection type D (15 cases in total; the resections of five patients involved mainly the vastus lateralis with type A, the resections of six patients involved mainly the vastus medialis with type B, and the resections of two patients involved the rectus femoris and vastus intermedius with type C. In one patient, resection was performed on the femoral nerve). ^j^ Four years of average follow-up in the 5 surviving patients with conservative surgical procedures (range: 51–163 months). The minimum follow-up period for four of these patients was 8 years. ^k^ Functional results of five patients who remained alive and retained their limb after a minimum four-year follow-up.

## Data Availability

The datasets used and/or analyzed during the current study are available from the corresponding author upon reasonable request.

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
