# Peer review of "Total Quadriceps Resection in High-Grade Soft-Tissue Sarcomas of the Thigh: Surgical Technique and Long-Term Functional Outcomes in Surviving Patients"

_cancers, 2025, doi:10.3390/cancers18010037_

Round 1
Reviewer 1 Report
Comments and Suggestions for Authors
The manuscript entitled "Total Quadriceps Resection in High-grade Soft Tissue Sarcomas of The Thigh. Long-term Functional Outcomes in Surviving Patients" aims to study outcomes after a minimum 4-year follow-up in surviving patients with limb preservation with high-grade soft-tissue sarcomas who required total quadricectomies followed by quadriceps reconstruction. The study is well conducted and analysed.
My only query is that the age of the patient varied from 35 to 79, it would be good to see the analysis based on the age of the patient. The complications in the patient based on the age should be included in a paragraph in the results and discussion section.
The conclusion section could be a bit more elaborate.
Author Response
QUESTION: My only query is that the age of the patient varied from 35 to 79, it would be good to see the analysis based on the age of the patient. The complications in the patient based on the age should be included in a paragraph in the results and discussion section.
RESPONSE:
This line has been added to the Results section: Six patients suffered necrosis or dehiscence of the wound margins. Five of them were older than 65 years.
This paragraph has been added to the Discussion section:
Despite the high rate of local wound complications in our series, the technique described is simple to perform and yields acceptable functional results. Active extension and strength were however significantly reduced, and this compromised knee stability. For all these reasons, the procedure could be more suitable for partial quadriceps resections [5], or in older low-demand patients or those with comorbidities not fit for prolonged microsurgical reconstruction. Nevertheless, particularly in older patients and according to our findings, postoperative necrosis of the surgical wound edges appears to be more frequent and should be taken into consideration. Similar results were reported in the series by Fisher et al. [8]. In their study of 43 patients undergoing hamstring transfer, postoperative complications occurred in sixteen patients (37%). Wound dehiscence was seen in 10 (62%), lymphedema in 2 (12%), fistula in 1 (6%) and hematoma in 1 patient (6%). Additionally, 2 patients developed both wound dehiscence and lymphedema (12%).
QUESTION: The conclusion section could be a bit more elaborate.
RESPONSE
The conclusion has been expanded and further elaborated: Total quadriceps resection in large high-grade soft-tissue sarcomas of the anterior compartment of the thigh ensures wide resection margins and local control of the disease, though local wound complications are common, particularly en older patients. Resection appears to be technically easier if performed distally to proximally in the thigh. Reconstruction can be performed using local muscle transfers in low-demand patients, while free neurotized muscle flaps are advisable in selected patients, mainly motivated young people. The anterolateral flap of the vastus lateralis muscle of the thigh is a good alternative to the latissimus dorsi flap, providing similar versatility while avoiding donor-site morbidity associated with harvesting a major muscle of the upper back.
Reviewer 2 Report
Comments and Suggestions for Authors
This manuscript presents a retrospective case series of ten patients undergoing total quadriceps resection for AJCC IIIB anterior thigh soft-tissue sarcomas, focusing on reconstructive strategies (local transfers vs. free functional muscle flaps) and long-term functional and oncologic outcomes. The topic is clinically relevant and addresses a challenging and underreported area, however, some points need to be addressed.
1- Express what this manuscript adds to the previously published data on quadricectomy.
2- The study is retrospective but lacks essential details such as: Period of data collection, Inclusion and exclusion criteria, How missing data or lost follow-up were handled, and whether all consecutive cases were included.
3- While assessing the MSTS scores, what was the exact time, and was the assessor blinded?
4- Functional results are presented only for survivors, creating survival bias. Acknowledge and quantify how excluding deceased early patients affects interpretation.
5- Some results (e.g., mean knee flexion) are reported only for 5 survivors. Provide results for the entire cohort where applicable or justify exclusion.
6- Please provide the Institutional Review Board (IRB) approval status.
Author Response
QUESTION 1: Express what this manuscript adds to the previously published data on quadricectomy.
RESPONSE:
Technically, quadricectomy is performed from distal to proximal, whereas it was traditionally recommended to be performed from proximal to distal. We consider that this straightforward technical maneuver simplifies the procedure and facilitates the resection.
With regard to reconstruction, given that there is no consensus on the best method, we propose a simple one for older patients and a more complex one for selected younger patients: the anterolateral flap of the vastus lateralis muscle of the thigh. In similar cases, the latissimus dorsi flap had been used most often, which has higher morbidity as it is a major muscle of the upper back. This aspect has been emphasised in the conclusion, which has been expanded.
As we say in the introduction section, the best reconstruction technique remains to be defined, and evidence is limited to brief case series. We have added a table clarifying this statement: see Table 3.
QUESTION 2: The study is retrospective but lacks essential details such as: Period of data collection, Inclusion and exclusion criteria, How missing data or lost follow-up were handled, and whether all consecutive cases were included.
RESPONSE:
Certainly, essential details of the study design were missing. Apologies. We have included them in the first paragraph of the Maerials and Methods section. We have also emphasised them in the paragraph on the limitations of the study, at the end of the discussion section.
In Materials and Methods: We performed a retrospective review of patients treated for AJCC stage IIIB high-grade soft-tissue sarcomas of the anterior compartment of the thigh treated in two hospitals, all but one by the same surgeon, from 2012 to 2025. Consecutive patients with the aforementioned diagnosis who underwent total quadriceps resection and suffered complete loss of the knee extensor mechanism were included in the study. Patients were excluded if longitudinal continuity of any of the quadriceps muscle complex was preserved. During follow-up, there was no loss of patients or of any data required for the study.
In Discussión: Despite these limitations, we believe our study fulfills its proposed objectives. The participation of a single surgeon in all surgeries supports the reliability of the results observed in this series, even though it is multicentric. Furthermore, although the study design was retrospective, there was no loss of patients during follow-up and all necessary data were recorded in the medical records.
QUESTION 3: While assessing the MSTS scores, what was the exact time, and was the assessor blinded?
RESPONSE:
The MSTS score was assessed by the principal investigator based on values recorded by a specialist in physical therapy and rehabilitative medicine in 7 of 10 patients who survived more than 3 years at the end of the study or at the last review before death (cases 1 and 9). The relevant paragraph in the method section specifies that: To avoid assessor bias of the primary operating surgeon, we used the values recorded by a specialist in physical therapy and rehabilitative medicine. In other words, the physiotherapist conducting the assessment was blinded to the study.
The full paragraph reads as follows: Functional results were evaluated according to the Musculoskeletal Tumor Society (MSTS) scale [16] in 7 of 10 patients who survived more than 3 years at the end of the study or at the last review before death (cases 1 and 9). MSTS scale converts the score allocated to six criteria into percentages, and considers the results above 79% good or excellent, between 60 and 79% acceptable, and poor below 60%. Strength was graded from 0 – 5 according to the Medical Research Council Manual Muscle Testing scale [17]. To avoid assessor bias of the primary operating surgeon, we used the values recorded by a specialist in physical therapy and rehabilitative medicine.
QUESTION 4: Functional results are presented only for survivors, creating survival bias. Acknowledge and quantify how excluding deceased early patients affects interpretation.
RESPONSE:
This is certainly the case, and there would be a survival bias. Although the results are detailed in Table 2, in the results and discussion sections we only refer to the five surviving patients who retained their limbs at the end of the study, as we consider them to be relevant for the purposes of assessing the patient's long-term functionality.
In the final paragraph of Materials and Methods, we clarify: Surgical complications (e.g., hematoma, infection, wound dehiscence), functional and oncological results of treatment were analyzed at final follow-up or death. Functional results were evaluated according to the Musculoskeletal Tumor Society (MSTS) scale [16] in 7 of 10 patients who survived more than 3 years at the end of the study or at the last review before death (cases 1 and 9).
This insightful observation is included in the limitations paragraph at the end of the discussion section: Second, oncologic and functional follow-up was reduced to seven cases, due to the death of two patients in the early postoperative period and early local recurrence that necessitated amputation in another case; nonetheless, all patients that were alive at the time of minimum follow-up could be evaluated and included in this series. Therefore, the survival bias that may exist with regard to functional outcomes is not relevant to the analysis of long-term limb functionality.
QUESTION 5: Some results (e.g., mean knee flexion) are reported only for 5 survivors. Provide results for the entire cohort where applicable or justify exclusion.
RESPONSE:
Knee flexion was recorded in seven patients, excluding those for whom this data was irrelevant. In two cases, this was because their survival was less than one year (cases 3 and 5) and in the patient who underwent amputation (case 10).
We justify the exclusion in the paragraph on limitations in the discussion section: Second, oncologic and functional follow-up was reduced to seven cases, due to the death of two patients in the early postoperative period and early local recurrence that necessitated amputation in another case; nonetheless, all patients that were alive at the time of minimum follow-up could be evaluated and included in this series. Therefore, the survival bias that may exist with regard to functional outcomes is not relevant to the analysis of long-term limb functionality. It is also irrelevant not to record the degree of knee flexion in patients with a survival rate of less than one year (cases 3 and 5) or who have undergone an amputation (case 10).
QUESTION 6: Please provide the Institutional Review Board (IRB) approval status.
RESPONSE:
The reference has been included in the materials and methods section: The study was submitted for consideration and approval by the ethics committee of our Center. All survivent patients provided written informed consent to be included in the case series. For non-surviving patient, data were irreversibly anonymized. All procedures were performed in accordance with the the Helsinki Declaration of 1975, as revised in 1983, as well as local legislation (Law 41/2002, regulating patient autonomy and the rights and obligations regarding information and clinical documentation).
Reviewer 3 Report
Comments and Suggestions for Authors
The authors' manuscript proposal presents a retrospective case series of 10 patients who underwent total quadriceps resection for grade IIIB sarcomas of the superficial muscular envelope and provided long term survival and rehabilitation outcomes. During the minimum follow-up was 4 years, 4 patients died, 1 required external hemipelvectomy due to disease recurrence, 5 were alive with intact limbs. All had required reoperation because of wound necrosis and other complications. Most patients were reconstructed with local muscle transfers, and two underwent free muscle flaps. As it is presented, this reviewer is unclear as to what new or confirmatory information this manuscript proposal is able to provide. Revising the present manuscript to be an individual participant data systematic review following the PRISMA guidelines (https://www.prisma-statement.org/ipd) that then contributes the additional 10 patients could make this a much stronger contribution to the literature. This reviewer looks forward to reading the revised manuscript proposal.
Comments on the Quality of English LanguageThe manuscript proposal is readable but periodically uses the wrong work or leaves words out that make reading tedious. Detailed proofreading, however, should remedy these issues. Professional services should not be needed for this.
In particular, this reviewer suggests replacing the period with colon in title.
Author Response
The authors' manuscript proposal presents a retrospective case series of 10 patients who underwent total quadriceps resection for grade IIIB sarcomas of the superficial muscular envelope and provided long term survival and rehabilitation outcomes. During the minimum follow-up was 4 years, 4 patients died, 1 required external hemipelvectomy due to disease recurrence, 5 were alive with intact limbs. All had required reoperation because of wound necrosis and other complications. Most patients were reconstructed with local muscle transfers, and two underwent free muscle flaps. As it is presented, this reviewer is unclear as to what new or confirmatory information this manuscript proposal is able to provide. Revising the present manuscript to be an individual participant data systematic review following the PRISMA guidelines (https://www.prisma-statement.org/ipd) that then contributes the additional 10 patients could make this a much stronger contribution to the literature. This reviewer looks forward to reading the revised manuscript proposal.
RESPONSE:
The new or confirmatory information in the manuscript refers to a simple modification of the classic resection technique that facilitates it and a proposal for the indication of the reconstructive method based on the age and condition of the patients. On the other hand, although partial quadriceps resections are relatively common, complete resection is rare, as demonstrated in the scientific literature (see new Table 3). The 10 cases in our series make it one of the largest published series to our knowledge.
About resection, technically quadricectomy is performed from distal to proximal, whereas it was traditionally recommended to be performed from proximal to distal. We consider that this straightforward technical maneuver simplifies the procedure and facilitates the resection. In addition to presenting it in Figure 3, we highlight it in the title modification, in the discussion, and in the conclusion of the manuscript.
Title: TOTAL QUADRICEPS RESECTION IN HIGH-GRADE SOFT TISSUE SARCOMAS OF THE THIGH: SURGICAL TECHNIQUE AND LONG-TERM FUNCTIONAL OUTCOMES IN SURVIVING PATIENTS.
Discussion: Unlike the classical technique [7,11], resection seems easier if the distal quadriceps tenotomy is performed first and the excision continued by pulling the muscle proximally.
Conclusions: Total quadriceps resection in large high-grade soft-tissue sarcomas of the anterior compartment of the thigh ensures wide resection margins and local control of the disease, though local wound complications are common, particularly en older patients. Resection appears to be technically easier if performed distally to proximally in the thigh. Reconstruction can be performed using local muscle transfers in low-demand patients, while free neurotized muscle flaps are advisable in selected patients, mainly motivated young people. The anterolateral flap of the vastus lateralis muscle of the thigh is a good alternative to the latissimus dorsi flap, providing similar versatility while avoiding donor-site morbidity associated with harvesting a major muscle of the upper back.
With regard to reconstruction, given that there is no consensus on the best method, we propose a simple one for older patients and a more complex one for selected younger patients: the anterolateral flap of the vastus lateralis muscle of the thigh. In similar cases, the latissimus dorsi flap had been used most often, which has higher morbidity as it is a major muscle of the upper back. This aspect has been emphasised in the conclusion, which has been expanded.
As we say in the introduction section, the best reconstruction technique remains to be defined, and evidence is limited to brief case series. We have added a table clarifying this statement: see Table 3. We believe that this table provides greater clarity than a systematic review due to the heterogeneity of studies on the subject. This is specified at the bottom of the new table:
Table 3. List of relevant publications on the reconstruction method following complete quadriceps resection and its functional outcomes. The selection was obtained through an electronic and manual search in Pubmed using the search string “soft tissue sarcoma AND quadriceps AND extensor AND reconstruction”. Among the 10 articles identified in the search, we excluded one referring to bone osteosarcomas reconstructed with megaprostheses [15], another on an approach to the distal femur through the knee extensor mechanism [9], another on a new technique of patellar ligament reconstruction [14], another on rotationplasty (Borggreve-Van Nes operation) [13], another that was an evidence-based review [10], and a fifth that was a letter to the editor [16]. This was replaced by the original article by Innocenti et al. [8], which it comments on. Three secondary references found via bibliographic linkage [7,12,19] were added. ALT = AnteroLateral Thigh. ISOLS = International Society Of Limb Salvage. FS = Fibrosarcoma. FU = Follow-up. LS = Liposarcoma. MFS = Myxofibrosarcoma. MSTS = Musculoskeletal Tumor Society. n = Number of cases. NS = Not Specified. OS = Osteosarcoma. QoL = Quality of Life (SF-36 questionnaire). ROM = Range Of Motion (degrees). RT = Radiotherapy. SS = Synovial Sarcoma. UPS = Undifferentiated Pleomorphic Sarcoma. aResection around the knee joint (quadriceps tendon, patella, and/or patellar tendon) without apparent joint involvement. bTotal quadriceps resection in 4 cases (11 cases in total). cAssociated surgery: sartorius transfer and massive allograft (1 case) or pedicled fascia lata (2 cases). dCalculated on 11 patients. eTotal quadriceps resection in 16 cases (37%), three-fourths in 12 cases (28%, vastus lateralis or medialis plus intermedius and rectus femoris muscle), half in 4 cases (9%, vastus lateralis/medialis and rectus femoris muscle) and the femoral nerve in 11 cases (26%). So, in 91% of cases 3/4 or more of the quadriceps muscle was removed. fResults for 9 patients available for physical examination with resection > 3/4 of the quadriceps. gTotal quadriceps resection in 2 patients (12 cases in total). hCalculated on the 12 patients in the series. iTotal quadriceps resection in 1 patient -resection type D- (15 cases in total: the resections of five patients involved mainly the vastus lateralis -type A-, the resections of six patients involved mainly the vastus medialis -type B-, and the resections of two patients involved the rectus femoris and vastus intermedius -type C-. In one patient, resection was performed on the femoral nerve).
In the discussion section, we reiterate this point with a new paragraph: Currently, consensus is lacking regarding the best method of reconstruction and controversy continues (Table 3). In fact, as Fisher et al (8) published, the selection of the surgical technique was not associated with the extent of quadriceps muscle resection but with the preference and experience of the surgeon.
QUESTION 2: Comments on the Quality of English Language
The manuscript proposal is readable but periodically uses the wrong work or leaves words out that make reading tedious. Detailed proofreading, however, should remedy these issues. Professional services should not be needed for this.
In particular, this reviewer suggests replacing the period with colon in title.
RESPONSE:
We tried to improve the wording of the manuscript. We corrected the suggestion in the title. Thank you very much.

Reviewer 4 Report
Comments and Suggestions for Authors
This is a well-prepared manuscript reporting on the functional outcomes of patients who underwent total quadriceps resection for high-grade soft-tissue sarcomas. The study provides detailed descriptions of ten rare and complex cases, offering valuable clinical insights for orthopedic oncologists. I would, however, suggest adding further clarification regarding the reconstruction methods applied in each case.
Specifically, it would greatly aid readers’ understanding if the type of reconstruction (e.g., local muscle transfer vs. free neurotized muscle flap) were explicitly indicated for each patient. The most straightforward and informative approach would be to include this information as an additional column in the summary table of cases.
Author Response
QUESTION: Specifically, it would greatly aid readers’ understanding if the type of reconstruction (e.g., local muscle transfer vs. free neurotized muscle flap) were explicitly indicated for each patient. The most straightforward and informative approach would be to include this information as an additional column in the summary table of cases.
RESPONSE:
We have added the column to Table 1, specifying more clearly the reconstruction method applied in each case. This correction certainly makes the paper easier to read and understand. Thank you for your observation.

Round 2
Reviewer 2 Report
Comments and Suggestions for Authors
Thank you
Author Response
QUESTION 1: Express what this manuscript adds to the previously published data on quadricectomy.
RESPONSE:
Technically, quadricectomy is performed from distal to proximal, whereas it was traditionally recommended to be performed from proximal to distal. We consider that this straightforward technical maneuver simplifies the procedure and facilitates the resection.
With regard to reconstruction, given that there is no consensus on the best method, we propose a simple one for older patients and a more complex one for selected younger patients: the anterolateral flap of the vastus lateralis muscle of the thigh. In similar cases, the latissimus dorsi flap had been used most often, which has higher morbidity as it is a major muscle of the upper back. This aspect has been emphasised in the conclusion, which has been expanded.
As we say in the introduction section, the best reconstruction technique remains to be defined, and evidence is limited to brief case series. We have added a table clarifying this statement: see Table 3.
QUESTION 2: The study is retrospective but lacks essential details such as: Period of data collection, Inclusion and exclusion criteria, How missing data or lost follow-up were handled, and whether all consecutive cases were included.
RESPONSE:
Certainly, essential details of the study design were missing. Apologies. We have included them in the first paragraph of the Maerials and Methods section. We have also emphasised them in the paragraph on the limitations of the study, at the end of the discussion section.
In Materials and Methods: We performed a retrospective review of patients treated for AJCC stage IIIB high-grade soft-tissue sarcomas of the anterior compartment of the thigh treated in two hospitals, all but one by the same surgeon, from 2012 to 2025. Consecutive patients with the aforementioned diagnosis who underwent total quadriceps resection and suffered complete loss of the knee extensor mechanism were included in the study. Patients were excluded if longitudinal continuity of any of the quadriceps muscle complex was preserved. During follow-up, there was no loss of patients or of any data required for the study.
In Discussión: Despite these limitations, we believe our study fulfills its proposed objectives. The participation of a single surgeon in all surgeries supports the reliability of the results observed in this series, even though it is multicentric. Furthermore, although the study design was retrospective, there was no loss of patients during follow-up and all necessary data were recorded in the medical records.
QUESTION 3: While assessing the MSTS scores, what was the exact time, and was the assessor blinded?
RESPONSE:
The MSTS score was assessed by the principal investigator based on values recorded by a specialist in physical therapy and rehabilitative medicine in 7 of 10 patients who survived more than 3 years at the end of the study or at the last review before death (cases 1 and 9). The relevant paragraph in the method section specifies that: To avoid assessor bias of the primary operating surgeon, we used the values recorded by a specialist in physical therapy and rehabilitative medicine. In other words, the physiotherapist conducting the assessment was blinded to the study.
The full paragraph reads as follows: Functional results were evaluated according to the Musculoskeletal Tumor Society (MSTS) scale [16] in 7 of 10 patients who survived more than 3 years at the end of the study or at the last review before death (cases 1 and 9). MSTS scale converts the score allocated to six criteria into percentages, and considers the results above 79% good or excellent, between 60 and 79% acceptable, and poor below 60%. Strength was graded from 0 – 5 according to the Medical Research Council Manual Muscle Testing scale [17]. To avoid assessor bias of the primary operating surgeon, we used the values recorded by a specialist in physical therapy and rehabilitative medicine.
QUESTION 4: Functional results are presented only for survivors, creating survival bias. Acknowledge and quantify how excluding deceased early patients affects interpretation.
RESPONSE:
This is certainly the case, and there would be a survival bias. Although the results are detailed in Table 2, in the results and discussion sections we only refer to the five surviving patients who retained their limbs at the end of the study, as we consider them to be relevant for the purposes of assessing the patient's long-term functionality.
In the final paragraph of Materials and Methods, we clarify: Surgical complications (e.g., hematoma, infection, wound dehiscence), functional and oncological results of treatment were analyzed at final follow-up or death. Functional results were evaluated according to the Musculoskeletal Tumor Society (MSTS) scale [16] in 7 of 10 patients who survived more than 3 years at the end of the study or at the last review before death (cases 1 and 9).
This insightful observation is included in the limitations paragraph at the end of the discussion section: Second, oncologic and functional follow-up was reduced to seven cases, due to the death of two patients in the early postoperative period and early local recurrence that necessitated amputation in another case; nonetheless, all patients that were alive at the time of minimum follow-up could be evaluated and included in this series. Therefore, the survival bias that may exist with regard to functional outcomes is not relevant to the analysis of long-term limb functionality.
QUESTION 5: Some results (e.g., mean knee flexion) are reported only for 5 survivors. Provide results for the entire cohort where applicable or justify exclusion.
RESPONSE:
Knee flexion was recorded in seven patients, excluding those for whom this data was irrelevant. In two cases, this was because their survival was less than one year (cases 3 and 5) and in the patient who underwent amputation (case 10).
We justify the exclusion in the paragraph on limitations in the discussion section: Second, oncologic and functional follow-up was reduced to seven cases, due to the death of two patients in the early postoperative period and early local recurrence that necessitated amputation in another case; nonetheless, all patients that were alive at the time of minimum follow-up could be evaluated and included in this series. Therefore, the survival bias that may exist with regard to functional outcomes is not relevant to the analysis of long-term limb functionality. It is also irrelevant not to record the degree of knee flexion in patients with a survival rate of less than one year (cases 3 and 5) or who have undergone an amputation (case 10).
QUESTION 6: Please provide the Institutional Review Board (IRB) approval status.
RESPONSE:
The reference has been included in the materials and methods section: The study was approved by the Ethics Committee (CEIm) of our Center, complying with patient anonymity requirements. All survivors provided written informed consent for publication. All survivent patients provided written informed consent to be included in the case series. For non-surviving patient, data were irreversibly anonymized. All procedures were performed in accordance with the the Helsinki Declaration of 1975, as revised in 1983, as well as local legislation (Law 41/2002, regulating patient autonomy and the rights and obligations regarding information and clinical documentation).
Reviewer 3 Report
Comments and Suggestions for Authors
This reviewer thanks the authors very much for their careful,
thoughtful, and responsive reply to the reviewer's comments. There are some improvements, and there are areas that require additional attention.
Much of the text associated with the new table 3 is misplaced. Part of it should have been in the Methods section, part in the Results, part in the Discussion.
This reviewer has published individual participant meta-analyses and regular systematic reviews and associated protocols. Making it an individual participant meta-analyses with contribution of the 10 patients in your series, highlighting the technique and indication modifications, would streamline the manuscript proposal and improve perceived objectivity by the reader. In the Discussion, the differences in studies can be explored.
As this reviewer sees it, the manuscript proposal's goal is twofold: 1- systematically highlight literature on the topic and 2- contribute 10 new cases illustrating a technique and indication modification and expected outcomes. To the reviewer, these points were unclear until reading the authors excellent response. It seems the manuscript proposal needs better streamlining and organization/structure.
This reviewer also specifically suggests to consider this a contribution of a case series and not a study. It is not really a study. There is no comparison. It is just prospective documentation of status, treatments, and outcomes. That does not make it worthless, but the authors need to be clear about what it is and what it is not. That will also help.
This reviewer hopes the above suggestions are helpful and looks forward to reading the next revision. Thank you for the opportunity to review this work.
English language concerns remain, mostly in word choice and spelling. Careful proofreading by a native, in-field English speaker is needed. This can be informal and should not require professional services.
Author Response
Comments: Much of the text associated with the new table 3 is misplaced. Part of it should have been in the Methods section, part in the Results, part in the Discussion.
This reviewer has published individual participant meta-analyses and regular systematic reviews and associated protocols. Making it an individual participant meta-analyses with contribution of the 10 patients in your series, highlighting the technique and indication modifications, would streamline the manuscript proposal and improve perceived objectivity by the reader. In the Discussion, the differences in studies can be explored.
Response: Table 3 aims to provide an overview of the few bibliographic references that exist on the subject and, in particular, on the technique. At the reviewer's suggestion, we added a new row to the table with our series of 10 cases to aid readers' understanding. We also emphasize this in the discussion section and in the footnote to the table: “List of relevant publications on the reconstruction method following complete quadriceps resection and its functional outcomes, including the results of our series for comparative purposes.”
We have added comments referring to Table 3 in the sections mentioned (Method, Results, and Discussion).
In the methods section, we have added the method for searching for publications on the topic, renumbering Table 3 as Table 1: “We performed a retrospective review of patients treated for AJCC stage IIIB high-grade soft-tissue sarcomas of the anterior compartment of the thigh treated in two hospitals, all but one by the same surgeon, from 2012 to 2025. Consecutive patients with the aforementioned diagnosis who underwent total quadriceps resection and suffered complete loss of the knee extensor mechanism were included in the study. Patients were excluded if longitudinal continuity of any of the quadriceps muscle complex was preserved. During follow-up, there was no loss of patients or of any data required for the study. We add a list of relevant publications on the reconstruction method following complete quadriceps resection and its functional outcomes (Table 1). The selection was obtained through an electronic and manual search in Pubmed using the search string “soft tissue sarcoma AND quadriceps AND extensor AND reconstruction”. Among the 10 articles identified in the search, we excluded one referring to bone osteosarcomas reconstructed with megaprostheses [15], another on an approach to the distal femur through the knee extensor mechanism [9], another on a new technique of patellar ligament reconstruction [14], another on rotationplasty (Borggreve-Van Nes operation) [13], another that was an evidence-based review [10], and a fifth that was a letter to the editor [16]. This was replaced by the original article by Innocenti et al. [8], which it comments on. Three secondary references found via bibliographic linkage [7,12,19] were added, as well as the results of our series for comparative purposes.”
ATTENTION: Table 3 has been renumbered as Table 1 and, consequently, the former Tables 1 and 2 are now Tables 2 and 3, respectively. For the same reason, the order in which the tables appear in the publication must be changed.
Comments: As this reviewer sees it, the manuscript proposal's goal is twofold: 1- systematically highlight literature on the topic and 2- contribute 10 new cases illustrating a technique and indication modification and expected outcomes. To the reviewer, these points were unclear until reading the authors excellent response. It seems the manuscript proposal needs better streamlining and organization/structure.
This reviewer also specifically suggests to consider this a contribution of a case series and not a study. It is not really a study. There is no comparison. It is just prospective documentation of status, treatments, and outcomes. That does not make it worthless, but the authors need to be clear about what it is and what it is not. That will also help.
Response: We agree that the objectives of the study are those indicated by the reviewer and that they have been clarified in the introduction and conclusions sections. Thank you for your comment.
We have replaced the term “study” with “case series” in the paragraph on the study objectives in the introduction section: “The aim of this case series is to present our outcomes after a minimum 4-year follow-up in surviving patients with limb preservation with high-grade soft-tissue sarcomas who required total quadricectomies followed by quadriceps reconstruction.”
We appreciate and are very grateful for the reviewers' comments, which we believe are significantly improving the manuscript. We look forward to hearing your comments.
Round 3
Reviewer 3 Report
Comments and Suggestions for Authors
The reviewer thanks the authors for their sincere efforts.
This manuscript proposal, however, would be immeasurably improved if the authors implemented PRISMA Guidelines. Their search description is insufficient. For example, the "search strategy they provide is, itself, incomplete. After a couple of attempts, I think it should Beas follows: ("sarcoma"[MeSH Terms] OR "sarcoma"[All Fields] OR ("soft"[All Fields] AND "tissue"[All Fields] AND "sarcoma"[All Fields]) OR "soft tissue sarcoma"[All Fields]) AND ("quadricep"[All Fields] OR "quadriceps muscle"[MeSH Terms] OR ("quadriceps"[All Fields] AND "muscle"[All Fields]) OR "quadriceps muscle"[All Fields] OR "quadriceps"[All Fields]) AND ("extensor"[All Fields] OR "extensores"[All Fields] OR "extensors"[All Fields]) AND ("plastic surgery procedures"[MeSH Terms] OR ("plastic"[All Fields] AND "surgery"[All Fields] AND "procedures"[All Fields]) OR "plastic surgery procedures"[All Fields] OR "reconstruction"[All Fields] OR "reconstructions"[All Fields] OR "reconstruct"[All Fields] OR "reconstructability"[All Fields] OR "reconstructable"[All Fields] OR "reconstructed"[All Fields] OR "reconstructible"[All Fields] OR "reconstructing"[All Fields] OR "reconstructional"[All Fields] OR "reconstructive"[All Fields] OR "reconstructs"[All Fields])
Unfortunately, the authors provide no other relevant methods details of their literature review. This impairs transparency and objectivity, two of the pillars of peer review publication.
If the authors can switch this to some version of a systematic review or meta-analysis of individual participant data following PRISMA Guidelines, with their case series added, I think this could be a very useful contribution to the literature. Other concerns this reviewer had have been improved.
Comments on the Quality of English LanguageEnglish language concerns remain, mostly in word choice and spelling. Careful proofreading by a native, in-field English speaker is needed. This can be informal and should not require professional services.